# Effects of Encircled Abdominal Compression Device in Colonoscopy: A Meta-Analysis

**DOI:** 10.3390/jcm9010011

**Published:** 2019-12-19

**Authors:** Toshihiro Nishizawa, Hidekazu Suzuki, Hajime Higuchi, Hirotoshi Ebinuma, Osamu Toyoshima

**Affiliations:** 1Gastroenterology, Toyoshima Endoscopy Clinic, Tokyo 157-0066, Japan; nisizawa@kf7.so-net.ne.jp (T.N.); t@ichou.com (O.T.); 2Department of Gastroenterology and Hepatology, International University of Health and Welfare, Mita Hospital, Tokyo 108-8329, Japan; higuchi-h@iuhw.ac.jp (H.H.); ebinuma@iuhw.ac.jp (H.E.); 3Department of Gastroenterology and Hepatology, Tokai University School of Medicine, Kanagawa 259-1193, Japan; 4Department of Gastroenterology, Graduate School of Medicine, The University of Tokyo, Tokyo 113-8655, Japan

**Keywords:** encircling abdominal compression device, colonoscopy, meta-analysis

## Abstract

Background and Aim: The efficacy of encircling abdominal compression devices in colonoscopies is inconsistent. We performed a meta-analysis of randomized controlled trials (RCTs) in which encircling abdominal compression devices were compared with control in colonoscopies. Methods: We systematically searched RCTs published in the Cochrane Library, PubMed, and the Igaku-Chuo-Zasshi database. The data from the eligible RCTs were combined using the random-effects model. The weighted mean differences (WMDs), pooled odds ratios (ORs), and 95% confidence intervals (CIs) were calculated. Results: Five RCTs were included in this meta-analysis. Compared to the control group, encircling abdominal compression devices significantly reduced the caecal intubation time (WMD: −1.31, 95% CI: −2.40 to −0.23, *p* = 0.02). Compared to the control group, encircling abdominal compression devices significantly decreased the frequency of postural change (OR 0.30, 95% CI: 0.22 to 0.41, *p* < 0.00001). Compared to the control group, the use of encircling abdominal compression devices significantly reduced the need for abdominal compression (OR: 0.35, 95% CI: 0.17 to 0.70, *p* = 0.003). Conclusions: Encircling abdominal compression devices in colonoscopies was found to reduce the caecal intubation time and the frequency of abdominal compression.

## 1. Introduction

Colonoscopy is essential for colorectal cancer screening as well as therapeutic procedures [1,2,3,4]. Despite the advancements in colonoscopy equipment and progress in the endoscopists’ skills, colonoscopy can be difficult and painful for some patients. The formation of a sigmoid loop causes severe pain and makes endoscope insertion difficult. To prevent sigmoid loop, ancillary maneuvres such as position change and abdominal pressure are often added. However, position change needs at least one or more assistants, when the patients are under sedation. Abdominal pressure depends on the assistants’ skill, and sometimes does not exert effective pressure when inappropriately applied. Alternatively, a corset to encircle the abdomen might exert a well-balanced and effective pressure for the entire duration of the colonoscopy (Figure 1). Several randomized controlled trials (RCTs) have investigated the efficacy of encircling abdominal compression devices in colonoscopies [5,6,7,8,9]. Toros et al. and Yu et al. showed that encircling abdominal compression devices significantly reduced the caecal intubation time [6,9]. However, Crockett et al. reported inconsistent results [5]. Recently, we conducted an RCT to determine the efficacy and safety of an encircling abdominal compression device (back brace support belt: Maxbelt^®^) [7]. The intubation time in the Maxbelt group was shorter than that in the control group, but the difference was not significant (3.29 vs. 4.49 min, *p* = 0.069). 

The sample size in several trials was too small to achieve statistically conclusive results. We considered that systematically pooling the data from all RCTs could provide a better understanding of the efficacy of encircling abdominal compression devices in colonoscopies. Our aim was to conduct a systematic review and meta-analysis of RCTs evaluating the efficacy of encircling abdominal compression devices in colonoscopies. 

## 2. Methods

### 2.1. Search Strategy

The Cochrane Library, PubMed, Web of Science and the Igaku-Chuo-Zasshi database in Japan (up to November 2019) were used in this systematic review [10]. The following words were used for the systematic literature search: (colonoscopy) and (corset or bandage or abdominal compression device). There were no restrictions on language. 

### 2.2. Inclusion and Exclusion Criteria

Inclusion criteria were listed as follows: (1) study type: RCT; (2) population: patients who underwent a colonoscopy; (3) intervention: encircling abdominal compression device; (4) comparator: no device; (5) outcome: safety and efficacy of an encircling abdominal compression device. 

Exclusion criteria were listed as follows: (1) pillow-type abdominal compression devices; (2) conference abstracts; (3) duplicate publications; (4) reviews. 

### 2.3. Outcome Measures

The primary outcome of this meta-analysis was caecal intubation time. The secondary outcomes were the frequency of postural change and abdominal compression, and patient-reported comfort levels on a 5-point scale (5 = severe discomfort, 4 = moderate discomfort, 3 = mild discomfort, 2 = minimal discomfort, 1 = no discomfort).

### 2.4. Data Extraction

The following data were extracted from each study: first author, publication year, country, characteristics of patients (number, age, and gender), details of encircling abdominal compression devices, drugs for sedation, and outcomes. Two reviewers (T.N. and H.H.) independently investigated all articles for eligibility. 

### 2.5. Assessment of Methodological Quality

The quality of the included literature was assessed using the risk-of-bias tool outlined in the Cochrane Handbook for Systematic Reviews of Interventions (version 5.1.0) [11]. Two reviewers (T.N. and H.S.) scrutinized all studies and confirmed 6 items for RCT quality assessment: sequence generation, allocation concealment, blinding of patients and outcome assessors, management of incomplete outcome data, completeness of outcome reporting, and other potential threats to validity. 

### 2.6. Statistical Analysis

Statistical analysis was conducted using the Review Manager (RevMan; The Cochrane Collaboration, 2008; The Nordic Cochrane Centre, Copenhagen, Denmark). We used a random-effects model and Mantel-Haenszel method to calculate the odds ratio (OR), and we used inverse variance for the continuous data to estimate the weighted mean difference (WMD) with 95% confidence intervals (CI) [12]. Heterogeneity in the results of the studies was assessed by Cochran’s Q and I^2^ tests. Because of the low power of the Q test, *p* values < 0.1 were considered significant for heterogeneity. I^2^ score of ≥50% was considered to indicate a moderate level of heterogeneity [13]. For the purpose of our analysis, the standard deviation (SD) was estimated from the interquartile range (SD = interquartile range × 0.74) [14]. Finally, we used funnel plot asymmetry to detect any publication bias in the meta-analysis and Egger’s regression test to measure funnel plot asymmetry [15,16,17].

## 3. Results

### 3.1. Search Results

The systematic review process yielded 26 citations (Figure 2). Among them, 19 studies were excluded according to the exclusion criteria (6 unrelated topics, 2 conference abstracts, 9 duplicates, 1 letter, and 1 case report). The remaining seven studies were scrutinized. Then, two more studies were rejected, because pillow-type abdominal compression devices were used. Finally, five RCTs were included in the systematic review and meta-analysis [6,9]. The main characteristics of the eligible RCTs are shown in Table 1.

### 3.2. Quality Assessment

The risk of bias summary is presented in Table 2. In general, the quality of the included RCTs was considered high, except for the Tsutsumi et al.’s study. Tsutsumi et al.’s study did not perform random sequence generation and allocation concealment. In one RCT, the patients were blinded, and in three RCTs, the outcomes assessment was blinded. All five RCTs were found to have adequately assessed the incomplete outcomes, avoided selective outcome reporting, and were free of other biases. 

### 3.3. Meta-Analysis Results 

#### Caecal Intubation Time 

The caecal intubation time was described in four studies. Compared to the control group, encircling abdominal compression devices significantly reduced the caecal intubation time (WMD: −1.31, 95% CI: −2.40 to −0.23, *p* = 0.02, Figure 3). However, there was also significant heterogeneity between studies (*p* = 0.0006, I^2^ = 83%). Although Toros et al. had no blinding of the outcome assessment; the other three studies incorporated blinding. When Toros et al.’s study was excluded, the heterogeneity ceased to exist (*p* = 0.12). However, the exclusion of Toros et al.’s study did not significantly alter the result of the meta-analysis. Egger’s regression test suggested no significant asymmetry of the funnel plot (*p* = 0.757), indicating no evidence of substantial publication bias (Figure 4).

### 3.4. Postural Change 

The postural change was described in four studies. Compared to the control group, the pooled OR of postural change was 0.30 (95% CI: 0.22 to 0.41, *p* < 0.00001), indicating a significant decrease in the need for postural change (Figure 5A). There was no significant heterogeneity between four studies (*p* = 0.39, I^2^ = 1%).

### 3.5. Abdominal Compression

The abdominal compression was described in four studies. Compared to the control group, the pooled OR of abdominal compression was 0.35 (95% CI: 0.17 to 0.70, *p* = 0.003), indicating a significant decrease in the frequency of abdominal compression required (Figure 5B). However, there was significant heterogeneity between four studies (*p* = 0.003, I^2^ = 79%,). Although Crockett et al. performed a colonoscopy under propofol sedation, three other studies performed a colonoscopy under midazolam or no sedation. When Crockett et al.’s study was excluded, the heterogeneity ceased to exist (*p* = 0.29). 

### 3.6. Patient-Reported Comfort Level

The patient-reported comfort level was described in five studies (Figure 5C). Compared to the control group, encircling abdominal compression devices did not significantly reduce the patient-reported comfort level (WMD: −0.58, 95% CI: −1.27 to 0.11, *p* = 0.10).

## 4. Discussion

Overall, we found that encircling abdominal compression devices significantly reduced the caecal intubation time and the frequency of postural change and abdominal compression.

Encircling abdominal compression devices could maintain the colonoscope in a straight position in the sigmoid colon and prevent looping during colonoscopy, thus facilitating a more comfortable insertion. Unlike manual abdominal pressure or position change, encircling abdominal compression devices generally provide effective pressure which facilitates insertion without the help of assistants [9].

Crockett et al. reported that the encircling abdominal compression device (ColoWrap^®^) did not decrease caecal intubation time [5]. However, the subgroup analysis revealed that ColoWrap^®^ significantly reduced the caecal intubation time in obese patients (body mass index of 30–40). Toyoshima et al. reported that the encircling the abdominal compression device (Maxbelt^®^) did not significantly decrease the caecal intubation time [7]. However, the subgroup analysis revealed that the Maxbelt^®^ significantly reduced the caecal intubation time in patients of higher age (age ≥ 45). Obesity and higher age are associated with a redundant colon [5,18]. These results suggest that abdominal pressure by abdominal compression devices might overcome looping associated with a redundant colon. 

Toyoshima et al. and Toros et al. described the prices of encircling abdominal compression devices. The prices were $16.20 per belt, and $10 per corset, respectively. The belt or corset can be reused. They can also be washed. Therefore, the cost of the devices could be quite low.

Several limitations of this meta-analysis should be addressed. Different methods for the blinding of outcomes assessment and sedation during a colonoscopy might be considered as a source of heterogeneity. The inter-study variability in the experience of the endoscopists could be also considered a source of heterogeneity. More experienced endoscopists would generally have less trouble managing loops and this would minimize any benefit using the devices. Furthermore, the subgroup analysis was not combined, because the cut-off lines of body mass index and age varied in different studies. There were no studies from Europe and there is a necessity for trials in European countries. Further studies with larger numbers of patients are warranted to understand the safety and efficacy of encircling abdominal compression devices in colonoscopies.

In conclusion, the use of encircling abdominal compression devices during colonoscopies was found to reduce the caecal intubation time and frequency of abdominal compression. However, the studies point to a minimal benefit compared to control which practically, may not be important enough to change practice.

## Figures and Tables

**Figure 1 jcm-09-00011-f001:**
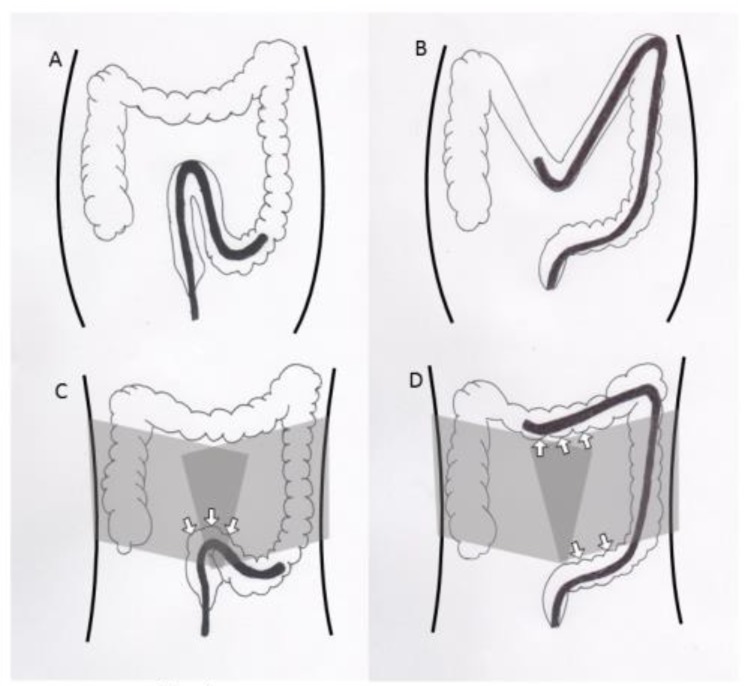
The effect of an encircled abdominal compression device (**A**) The elongation of a sigmoid colon without an encircled abdominal compression device. (**B**) The elongation of a transverse colon without the compression device. (**C**) Passing of a sigmoid colon with the compression device. Arrows indicate the compression for a sigmoid colon. (**D**) Passing of a transverse colon with the compression device. Arrows indicate the compression for a transverse and sigmoid colon.

**Figure 2 jcm-09-00011-f002:**
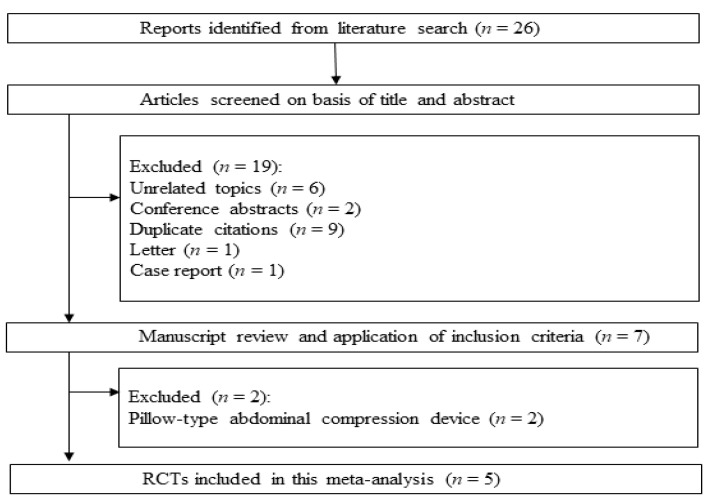
Flow diagram of the literature search and selection process. RCTs: randomized controlled trials.

**Figure 3 jcm-09-00011-f003:**
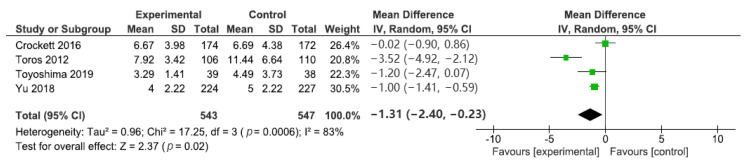
Forest plot of the weighted mean difference and 95% confidence intervals (CI) for the caecal intubation time in colonoscopy. df: degrees of freedom.

**Figure 4 jcm-09-00011-f004:**
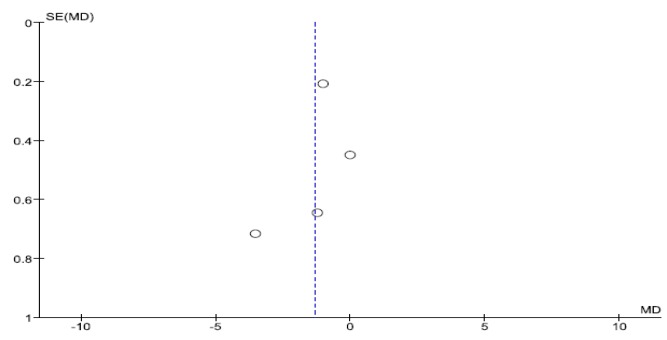
Funnel plot of the included studies for cecal intubation time in colonoscopy.

**Figure 5 jcm-09-00011-f005:**
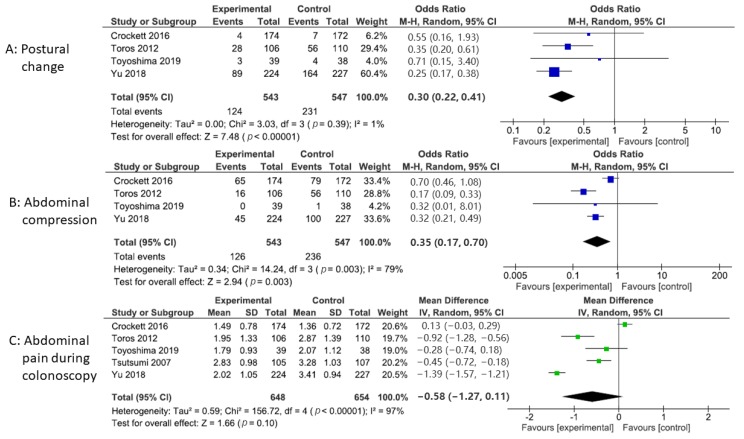
(**A**) Forest plot of the odds ratio for postural change in colonoscopy. (**B**) Forest plot of the odds ratio for abdominal compression in colonoscopy. (**C**) Forest plot of the weighted mean difference for abdominal pain during colonoscopy.

**Table 1 jcm-09-00011-t001:** Characteristics of studies included in the systematic review.

Author	Country	Abdominal	Sedation	Allocation	Patients	Age	Gender
Year		Compression Device			Number	±SD	M/F
Crockett	USA	Abdominal wrap	Propofol	Wrap with strap	175	59.9 ± 8.7	67/108
2016		with strap (ColoWrap^®^)		Sham	175	61.1 ± 8.1	67/108
Toros	Turkey	Abdominal	None	Corset	105	43.1 ± 13.1	49/57
2012		corset		Control	107	43.8 ± 13.5	48/62
Toyoshima	Japan	Back brace support	Midazolam	Belt	39	51.3 ± 10.1	24/15
2019		belt (Maxbelt^®^)	± Pethidine	Control	38	56.2 ± 11.5	25/13
Tsutsumi	Japan	Abdominal	None	Bandage	105	67.2 (18–87)	69/36
2007		bandage		Control	107	69 (20–84)	74/33
Yu	China	Abdominal	None	Binder	224	54.5 ± 13.4	95/129
2018		obstetric binder		Control	227	56.9 ± 13.0	86/141

**Table 2 jcm-09-00011-t002:** Evaluation of bias of randomized controlled trials (RCTs) included in the systematic review.

First	Random Sequence	Allocation	Blinding of Participants	Blinding of Outcome	Adequate Assessment	Selective Reporting	No Other
Author	Generation	Concealment	and Personnel	Assessment	of Incomplete Outcome	Avoided	Bias
Crockett	Yes	Yes	Yes	Yes	Yes	Yes	Yes
Toros	Yes	Yes	No	No	Yes	Yes	Yes
Toyoshima	Yes	Yes	No	Yes	Yes	Yes	Yes
Tsutsumi	No	No	No	No	Yes	Yes	Yes
Yu	Yes	Yes	No	Yes	Yes	Yes	Yes

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
