# Peer review of "Effects of Encircled Abdominal Compression Device in Colonoscopy: A Meta-Analysis"

_jcm, 2019, doi:10.3390/jcm9010011_

Round 1

Reviewer 1 Report

Congratulations on a well composed meta-analysis. 

The paper is well written with sound methodology. 

I enjoyed reading it but had a few questions that I felt should be addressed if possible in the review. 

The topic of abdominal compression device is interesting and I feel the studies point to a minimal benefit compared to control which practically may not be important enough to change practice. This could be mentioned in the discussion/conclusion. 

The other practical things are cost and actual time benefit. I am not sure the cost of such devices could be shared in the paper. Also, if the main outcome is 1.5 minute less on insertion, this has to be relatively weighed.  Some of the mean insertion times were as low as 3 min in Toyoshima study, and 11 min in the Toros study.  This inter-study variability is perhaps more important to highlight and explore.  

Some factors may include the experience of the endoscopists, or patient factors like age (interestingly, the Toros paper had the youngest group of patients).  More experienced endoscopists would generally have less trouble managing loops and this would minimize any benefit using the devices. The sub-group analysis did bring up some potential groups to explore this further in like older and obese patients. The other major factor is sedation in these patients. It is mentioned that the sedation was different in 1 study using Propofol, than the others. The amount of sedation may also effect whether an endoscopist would desire to change posture during the procedure. 

Lastly, intra-procedure postural change is not always a negative outcome. Many endoscopists do colonoscopy with programmed postural changes on insertion, and on withdrawal (for example in  doi: 10.1016/j.gie.2015.01.035).  

Overall, I think the paper is very good and publishable. Some of the suggestions above may help to enhance the understanding and applicability of the findings. 

Best regards, 

Author Response

The topic of abdominal compression device is interesting and I feel the studies point to a minimal benefit compared to control which practically may not be important enough to change practice. This could be mentioned in the discussion/conclusion.

Thank you very much. Following sentence was added into conclusion section.

“However, the studies point to a minimal benefit compared to control which practically may not be important enough to change practice.”

The other practical things are cost and actual time benefit. I am not sure the cost of such devices could be shared in the paper.

Although the cost of the devices was not described in three studies, Toyoshima et al. and Toros et al. described the prices of the devices. The prices were $16.2 per belt, and $10 per corset, respectively. The belt or corset can be reused. They can also be washed. Therefore, the cost of encircling abdominal compression devices could be quite low. These points were added into the revised manuscript.

 Also, if the main outcome is 1.5 minute less on insertion, this has to be relatively weighed.  Some of the mean insertion times were as low as 3 min in Toyoshima study, and 11 min in the Toros study.  This inter-study variability is perhaps more important to highlight and explore. 

Some factors may include the experience of the endoscopists, or patient factors like age (interestingly, the Toros paper had the youngest group of patients).  More experienced endoscopists would generally have less trouble managing loops and this would minimize any benefit using the devices.

Following sentences were added into limitation section. Thank you very much.

The inter-study variability in experience of the endoscopists could be also considered a source of heterogeneity. More experienced endoscopists would generally have less trouble managing loops and this would minimize any benefit using the devices.

 The sub-group analysis did bring up some potential groups to explore this further in like older and obese patients.

The subgroup analysis was performed in Crockett et al.’s RCT and Toyoshima et al.’s RCT. The cut-of lines of BMI and age were 30 kg/m2 and 60 years old in Crockett et al.’s RCT, and 25 kg/m2 and 45 years old in Toyoshima et al.’s RCT, respectively. Therefore, it was difficult to combine the subgroup analyses. Following sentences were added into limitation section.

Furthermore, subgroup analysis was not combined, because the cut-of lines of body mass index and age varied in different studies.

The other major factor is sedation in these patients. It is mentioned that the sedation was different in 1 study using Propofol, than the others. The amount of sedation may also effect whether an endoscopist would desire to change posture during the procedure. Lastly, intra-procedure postural change is not always a negative outcome. Many endoscopists do colonoscopy with programmed postural changes on insertion, and on withdrawal (for example in  doi: 10.1016/j.gie.2015.01.035). 

Thank you very much. “Reduced frequency of postural change” was deleted in conclusion section.

Reviewer 2 Report

This is a well performed systematic review on external compression devices in colonoscopy. The results appear to be clearly in favor of the use of compression devices.

There is however no mention on the cost of the devices and the cost should be discussed, as this would increase a cost for every procedure. Maybe cost calculations were performed in any of the studies.

This review improves the current literature

Author Response

This is a well performed systematic review on external compression devices in colonoscopy. The results appear to be clearly in favor of the use of compression devices.

There is however no mention on the cost of the devices and the cost should be discussed, as this would increase a cost for every procedure. Maybe cost calculations were performed in any of the studies.

This review improves the current literature.

Although the cost of the devices was not described in three studies, Toyoshima et al. and Toros et al. described the prices the devices. The prices were $16.2 per belt, and $10 per corset, respectively. The belt or corset can be reused. They can also be washed. Therefore, the cost of encircling abdominal compression devices could be quite low. These points were added into the revised manuscript.